# Persistent Asthma at School Age: Associated Factors in Preschool Children with Asthma

**DOI:** 10.3390/children10010033

**Published:** 2022-12-24

**Authors:** Kantisa Sirianansopa, Kanokpan Ruangnapa, Pharsai Prasertsan, Kantara Saelim, Wanaporn Anuntaseree

**Affiliations:** Division of Pulmonology, Department of Pediatrics, Faculty of Medicine, Prince of Songkla University, 15 Karnjanavanich Road, Hatyai 90110, Songkhla, Thailand

**Keywords:** preschool asthma, persistent asthma, environmental risk factors, protective factors, modifiable risk factors

## Abstract

Most patients with childhood asthma present their first symptoms at preschool age. Identifying modifiable risks and protective factors at an early age may help develop asthma prevention and control strategies. This study aimed to identify factors at preschool age that are associated with persistent asthma at school age. This retrospective observational study included preschool children with asthma from 2015 to 2020 at a university hospital in Southern Thailand. In total, 189 eligible participants (70.9% boys; median age, 7.6 [6.7, 8.5] years) were included. Wheeze characteristics included early transient wheeze, persistent wheeze, and late-onset wheeze that accounted for 55%, 27.5%, and 19.5% of the patients, respectively. Approximately 20% of the participants had persistent asthma. Breastfeeding was a protective factor (odds ratio [OR] 0.4 [0.2, 0.9], *p* = 0.04). The modifiable risk factors were siblings living in the same household (OR 2.6 [1.1, 6.2], *p* = 0.02) and residence in an industrial area (OR 3.8 [1.4, 10.5], *p* = 0.009). Additionally, presence of allergic rhinitis was associated with an increased risk of persistent asthma at school age (OR 3.6 [1.6, 8.2], *p* = 0.002). Early therapeutic interventions targeting modifiable factors provide a window of opportunity to prevent persistent asthma at school age.

## 1. Introduction

A great proportion of preschool children with asthma undergo symptom remission by school age [1]. A classification of wheezing phenotypes has been proposed to aid identification of individual risks and planning of phenotype-directed therapy. Three phenotypes exhibit a common pattern and have been stratified based on the age of onset and clinical remission at the age of six years (i.e., early transient wheeze, late-onset wheeze, and persistent wheeze) [2]. Frequent symptom exacerbations result in socioeconomic consequences that negatively impact the quality of life of the affected household and lead to parental stress [3]. The primary goals of asthma management in preschool children are the maintenance of clinical control and reduction of the future risk of irreversible airway disease. 

However, phenotype-directed therapies reflect disease severity and vary over time. The early identification of risk factors for persistent asthma could be a key element in disease management through prompt prediction of childhood asthma. Birth cohort studies have found evidence regarding both modifiable and unmodifiable factors associated with asthma development. Prenatal (e.g., parental asthma, parental atopy, and maternal smoking), perinatal (e.g., exposure to indoor mold and perinatal oxygen use), postnatal (e.g., allergic sensitization in early life), and environmental factors (e.g., smoking and outdoor pollution) are associated with childhood asthma [4]. The asthma predictive index is a contemporary tool that helps manage preschool asthma in current clinical practice [5]. Among the risk factors that comprise this model are unmodifiable risk factors, such as parental asthma, atopic disease, and allergic sensitization in early life. Accordingly, persistence asthma in childhood is driven by gene-environmental interaction (e.g., genetic, epigenetic, environment). Multifaceted modifiable environmental risk factors play a crucial role in the preventive strategies, but the essential components have not clearly determined. A focusing on allergen sensitization, pollutants, and nutrition may be influenced in the persistent asthma symptoms in later childhood. Breastfeeding and maintain sufficient levels of vitamin D during pregnancy through delivery decreased the risk of recurrent wheezing in early life, but the evidence of a lesser chance of persistent asthma at the age 6 years remains unclear [6,7]. Sensitization to common aeroallergens (e.g., house dust mite, indoor pets, mold in home environment) appear to be a more complex relation between indoor exposure and asthma outcome [8,9]. A meta-analysis found intervention designed to prevent exposure to a single aeroallergen did not significantly affect asthma development, but multifaceted interventions of aeroallergen avoidance demonstrated an importance protective effect both before and after the age of 6 years [10]. Additionally, maternal smoking during pregnancy and after birth had its strongest effect on young children, whereas environmental tobacco smoke exposure seemed relevant to asthma development in older children [11]. Furthermore, exposure to outdoor pollutants such as traffic-related pollution, living in an industrial area may be attributable to an increased risk of asthma at any age [12].

Identifying modifiable risk factors for childhood asthma can be particularly useful for developing preventive strategies against asthma at an early age. Modification of relevant risk factors might offer a window of opportunity to optimize asthma management in preschool children. This study aimed to identify factors associated with persistent asthma at school age in preschool children. Therapeutic interventions targeting modifiable risk factors and the promotion of protective factors against asthma could reasonably reduce asthma morbidities, such as school absences, emergency department visits, and hospitalizations.

## 2. Materials and Methods

This was a retrospective observational study targeting preschool asthmatic children at the Pediatric Outpatient Clinic. This study was conducted at Songklanagarind Hospital, a university-affiliated teaching hospital, Thailand. The Human Research Ethics Committee Faculty of Medicine, Prince of Songkla University, Thailand (REC 65-059-1-1) approved the study with a waiver of inform consent. The ethic approval date was 28 February 2022. Patient information was extracted from the medical records based on diagnosis according to the International Classification of Diseases, 10th revision codes: R06.2 (wheezing) and J45 (asthma) between 2015 and 2020. The inclusion criteria were: (1) children ≤ 5 years, diagnosed with asthma, and prescribed inhaled corticosteroids (ICS) and (2) regular follow-up until six years of age. Patients with following criteria were excluded; (1) pre-existing lung disease such as congenital airway anomalies (e.g., vascular rings or bronchial atresia), bronchopulmonary dysplasia (BPD) or chronic lung disease, and (2) incomplete medical records regarding important data (i.e., family history, perinatal history, environmental variables, comorbidities, compliance, and medication).

### 2.1. Data Collection

Demographic data was extracted from the medical records and included sex, mode of delivery (vaginal or cesarean birth), age at first presentation of wheezing, perinatal oxygen use, breastfeeding at the age six months, history of parents and/or siblings asthma, exposure to environmental hazards before six years of age (residence in an industrial area, exposure to second hand smoke), indoor pets, day care attendance, presence of one or more children in the same household (siblings living in the same household), history of respiratory syncytial virus (RSV) infection before the onset of the first wheeze, the occurrence of epidemics of exacerbation related to climate change from July to November (seasonal wheezing), wheezing characteristics (early transient wheezing, late-onset wheezing, or persistent wheezing), comorbidities of allergic rhinitis and/or atopic dermatitis, absolute blood eosinophil count before prescription of ICS therapy, and the skin prick test results at any age.

### 2.2. Outcome Measurement

Preschool children with asthma having regular follow-up data until the age of six years (school age) were classified into two categories based on ICS prescription: (1) discontinuation of ICS therapy without asthma symptoms and no further wheezing exacerbation for at least one year of monitoring (non-asthma at school age group), and (2) continuous ICS prescription to maintain symptom control (asthma at school age group). The primary endpoint was the determination of potential factors associated with persistent asthma at school age. The secondary outcome was the prevalence of persistent asthma from preschool to school age. 

### 2.3. Operational Definitions

Preschool asthma [13]: a history of recurrent wheezing (≥3 times/year) in children ≤5 years of age with the following: (1) wheezing or coughing that occurs with exercise, laughing, or crying, in the absence of respiratory infection; (2) a history of allergic disease (atopic dermatitis or allergic rhinitis) or asthma in first-degree relatives; (3) clinical improvement during 3 months of ICS treatment; and worsening after cessation. Allergic sensitization to aeroallergens: a positive skin prick test for at least one inhalant-allergen Perinatal oxygen use: required oxygen therapy within seven days of birth. Residence in an industrial area: presence of an industrial factory or other similar establishments within 5 km^2^ of the child’s residence. Early transient wheeze: wheezing occurred before the age of 3 years and resolved by the age of six years. Late-onset wheeze: wheezing developed after 3 years of age and usually persists until 6 years of age. Persistent wheeze: wheezing started before 3 years of age and persisted throughout childhood. Atopic disease: physician’s diagnosis of allergic rhinitis and/or atopic dermatitis present in the medical records. Symptom exacerbation: presented with an acute or subacute increase in wheezing, shortness of breath, exercise intolerance, and coughing, especially while asleep resulting in impairment of daily activities. Good drug compliance: ICS use duration of >80% of the total treatment period.

### 2.4. Statistical Analysis

All statistical analyses were performed using R software, Version 4.1.2 (R Foundation for Statistical Computing, Vienna, Austria). Categorical variables (sex, breastfeeding, mode of delivery, perinatal oxygen use, parental asthma, age at first presentation of wheezing, blood eosinophil count, environmental exposure, allergic rhinitis, and atopic dermatitis) were presented as frequency and percentage and compared using a chi-square test or Fisher’s exact test. The Shapiro–Wilk test was used to check the normality of continuous variables. Nonparametric continuous variables were presented as median (interquartile range [IQR]). The Wilcoxon rank-sum test with continuity correction was used with non-normally distributed continuous variables. Univariate analyses were performed to identify the factors associated with persistent asthma at the age of 6 years. Independent variables with *p* < 0.2 in the univariate analysis were included in the multivariate model. The multivariate model with the lowest Akaike information criteria was judged as the most parsimonious model. *p* < 0.05 indicated a statistical significance.

## 3. Results

The initial review included 230 medical records; finally, 189 preschool children with asthma were eligible for analysis (41 met the exclusion criteria: incomplete data [n = 39] and BPD [n = 2]). Data revealed that 20% of preschool children with asthma had persistent symptoms that required ICS therapy until school age. Half of the preschool children with asthma had their first episode of wheezing within the first year of life. Characteristics of wheezing comprised: early transient wheeze (55%), persistent wheeze (27.5%), and late-onset wheeze (19.5%). During the study period, the primary caregiver was the mother (88.4%), and majority of the children were breast-fed (66.4%). Most patients had good drug compliance (82%). The median interquartile range of number of exacerbations during ICS therapy could be categorized as emergency department visits: 2 (1, 3) times per year and hospitalization: 1 (1, 2) times per year. The patient characteristics included male sex (70.9%), born at term via cesarean section (60.8%), perinatal oxygen use (7.4%), parental asthma (15.3%), and siblings diagnosed with asthma (11.1%). Additional comorbidities were allergic rhinitis (40%) and atopic dermatitis (14%). Smoking exposure (38.1%), day care attendance (28.6%), indoor pets (14.8%), and residence in an industrial area (12.7%) were identified as other predisposing factors. A history of RSV infection before the onset of the first wheezing event was found in 23.8% of the children. The factors associated with persistent asthma at the age of 6 years, identified by univariate analysis, are shown in Table 1. The following covariates were significantly different between the groups: breastfeeding, perinatal oxygen use, siblings diagnosed with asthma, residence in an industrial area, siblings living in the same household, and allergic rhinitis. Multivariate analysis of the independent factors associated with persistent asthma at the age of 6 years is presented in Figure 1. Children who were breast-fed were less likely to have persistent asthma at school age than the bottle-fed group (odds ratio [OR] 0.4 [0.2, 0.9], *p* = 0.04). The predisposing factors that carried a significantly higher risk were siblings living in the same household (OR 2.6 [1.1, 6.2], *p* = 0.02) and residence in an industrial area (OR 3.8 (1.4, 10.5), *p* = 0.009). Moreover, having a personal history of allergic rhinitis was associated with a significantly increased risk of persistent asthma at 6 years of age (OR 3.6 [1.6, 8.2], *p* = 0.002). Additionally, the factors of atopic dermatitis and aeroallergen sensitization included in the analysis; they appeared to increase the risk of persistent asthma, although not at a statistically significant level. 

## 4. Discussion

Our study determined the prevalence of persistent asthma in preschoolers until school age to be 20%. This study identified some modifiable risk factors and protective factors at preschool age that are predictive of persistent asthma symptoms in school-aged children, which could be useful in designing interventions to attain asthma management goals. We found potential modifiable factors related to both indoor (siblings living in the same household) and outdoor (residence in an industrial area) exposure. Furthermore, allergic rhinitis was identified as a treatable risk factor for persistent asthma. Additionally, breastfeeding has been found to be a protective factor against persistent asthma in early childhood. 

Earlier longitudinal studies have demonstrated the prevalence of persistent asthma in preschoolers through adolescence to vary between 20 and 30%, depending on ethnicity and criteria of remission [14]. The present study defined clinical remission as being symptom-free and without controller medications after the age of 6 years to distinguish preschoolers with persistent asthma through school age. Compared with previous studies, we have shown a lower prevalence of persistent asthma from preschool age through adolescence, which may be attributed to our participants having good drug compliance and fewer exacerbations. The limited exacerbation and maintenance of symptom control in childhood are predictors of asthma remission in adolescence [15]. Allergic sensitization (allergic rhinitis and atopic dermatitis) in early life have long been recognized as risk factors for asthma development [16]. Regarding immature T-helper (Th) cell function in young children, the Th cell type 2 (Th2) was mainly driven as a component of immune response. A prolonged enhancing Th2 immunity (e.g., prenatal influences by maternal allergen exposure during pregnancy, external environmental factors triggering at the postnatal period) can result in airway inflammation and immunopathological changes at a sensible stage of lung development. These immune dysregulation between T-cell subpopulations at early age leading to a part of the complex pathogenesis of childhood asthma [17]. More than half of the children with asthma have symptoms of allergic rhinitis, which may crucially impact asthma control. Similarly, our data suggest that preschool asthma with allergic rhinitis have an increased risk of persistent asthma in later childhood. Therefore, treatment of this condition may improve asthma outcomes in children. We also identified two modifiable risk factors to be avoided: (1) residence in an industrial area, which supports the association between exposure to air pollutants and poor asthma control in children with asthma [18]. However, the magnitude of the effect is not uniform across residential settings and critical exposure times. Our data suggest that a residential setting within 5 km^2^ of an industrial factory is associated with persistent asthma in later childhood. Moreover, early-life exposure to ambient particulate matter <2.5 μm (PM 2.5) at an average concentration of 7.8 μm /m^3^ is reportedly associated with increased asthma exacerbation [19]. Therefore, actions to prevent the exposure of, particularly, young individuals to air pollutants should be compulsory at the individual, local, and national levels, as it represents a potentially modifiable risk factor for asthma prevention and the maintenance of good lung function. (2) The presence of one or more siblings within the same household was a modifiable factor for persistent asthma through school age. The proposed hypothesis involves that increasing exposure to other children may place younger children at an elevated risk for contracting wheezing-associated respiratory infections, resulting in early lung function restriction linked to persistent wheezing at school age [20]. A recent systematic review delineated that having siblings is a time-varying risk factor that may increase the risk of recurrent wheezing from preschool to school age, but not necessarily specific to subsequent asthma in adulthood [21]. Therefore, education related to personal hand hygiene and self-quarantine during viral illness could prove helpful in preventing the spread of episodic viral-induced respiratory illness and the exacerbation of wheezing among children living within the same household. In addition to determining the risk factors, we also identified breastfeeding as a protective factor for persistent asthma; breastfeeding may enhance the child’s immune response through the immune-modulating properties of breast milk, providing a protective effect against childhood asthma [22].

The strength of our study lies in the identification of two preventable risk factors that can aid early life detection and therapeutic interventions in preschoolers for reduced chances of persistent asthma in later childhood. Furthermore, during the study period, children had high drug compliance (>80%), resulting in fewer exacerbations and hospitalizations. There are some limitations to this retrospective study. First, spirometry was not performed to confirm remission due to limited performance of the children; we considered clinical remission by the median time of 3-year symptom monitoring, which may be enough to guarantee disease remission. Second, the proportion of preschool children with asthma having records of atopic dermatitis and skin prick test results was small; therefore, the relevant data lacked the power to demonstrate any difference. Third, a definitive correlation with specific inhaled chemical agents was not established owing to the lack of recorded data regarding the types of industry involved. Although no single exposure appears to be responsible for the development of asthma or its associated symptoms, a recent systematic review reported compelling evidence regarding the association of aromatic and aliphatic compounds with increased asthma symptoms [23]. Additionally, oxygen therapy during the perinatal period carry a high risk of childhood asthma, especially in premature infants [24]. Our study included children defined as full-term infants requiring oxygen therapy during the perinatal period to demonstrate this effect on matured lungs. Nevertheless, perinatal oxygen use was linked to an increased risk for persistent asthma at school age; however, this was not statistically significant. Further data focusing on the full-term infants with perinatal oxygen use could be explored to reveal this consequence.

Prompt modification of such risk factors at an early age could decrease the incidence of persistent asthma in later childhood. The treatment of comorbidities can also lead to better asthma outcomes. Further studies will include the potential variables such as spirometry parameters for confirming the disease remission and immunological data to determine the impact of innate and cell-mediated immunity in asthmatic children. We should pay special attention to the time-varying effect of modifiable risk factors for understanding wheeze trajectories, presenting an opportunity for early secondary intervention, especially those targeting relevant environmental factors. This could substantially minimize the risk of future asthma development in young children, which is of crucial importance in public health.

## 5. Conclusions

Exposure to air pollutants and the presence of siblings in the same household at an early age are modifiable risk factors for persistent asthma at school age. Unless impractical due to specific reasons, breastfeeding should be considered the standard infant feeding modality, given its ability to promote the child’s immunity and offer protection against asthma. The early identification and treatment of modifiable risk factors may help prevent subsequent asthma in later childhood.

## Figures and Tables

**Figure 1 children-10-00033-f001:**
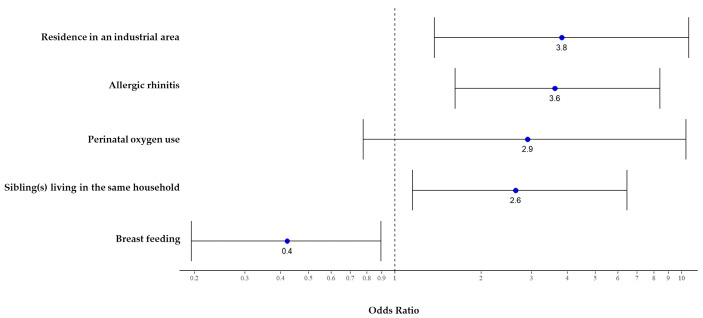
Odds ratios and 95% confidence intervals of the association between potential risk and protective factors and asthma outcomes at the age of six years.

**Table 1 children-10-00033-t001:** Factors associated with persistent asthma at school age (n = 189).

Characteristics	Asthma at School Age Group (n = 37)	Non-Asthma at School Age Group (n = 152)	Preschool Children with Asthma (189)	*p*-Value
Male sex	27 (73)	107 (70.4)	134 (70.9)	0.91
**Perinatal and post-natal factors**				
Route of delivery				1
Vaginal delivery	14 (37.8)	60 (39.5)	74 (39.2)	
Cesarean section	23 (62.2)	92 (60.5)	115 (60.8)	
Breast feeding	16 (43.2)	100 (65.8)	116 (61.4)	0.02 *
Perinatal oxygen use	6 (16.2)	8 (5.3)	14 (7.4)	0.03 *
**Family history**				
Parental asthma				0.59
Father	2 (5.4)	13 (8.6)	15 (7.9)	
Mother	4 (10.8)	10 (6.6)	14 (7.4)	
Sibling with asthma	8 (21.6)	13 (8.6)	21 (11.1)	0.04 *
**Environmental exposure**				
Residence in an industrial area	11 (29.7)	13 (8.6)	24 (12.7)	0.002 *
Smoking exposure	15 (40.5)	57 (37.5)	72 (38.1)	0.88
Seasonal wheezing	25 (67.6)	99 (65.1)	124 (65.6)	0.93
Indoor pets	7 (18.9)	21 (13.8)	28 (14.8)	0.59
Daycare attendance	7 (18.9)	47 (30.9)	54 (28.6)	0.21
Sibling(s) living in the same household	26 (70.3)	75 (49.3)	101 (53.4)	0.03 *
History of respiratory syncytial virus infection	8 (66.7)	37 (77.1)	45 (75)	0.47
**Wheezing characteristics**				
Age at onset of first wheeze				0.53
Age ≤ 12 months	17 (45.9)	79 (52)	96 (50.8)	
Age 12–36 months	15 (40.5)	61 (40.1)	76 (40.2)	
Age > 36 months	5 (13.5)	12 (7.9)	17 (9)	
**Comorbidities**				
Allergic rhinitis	25 (67.6)	51 (33.6)	76 (40.2)	<0.001 *
Atopic dermatitis	9 (24.3)	18 (11.8)	27 (14.3)	0.09
† **Blood eosinophil count >500 cells/mm^3^**	13 (50)	36 (34.3)	49 (37.4)	0.21
‡ **Sensitization to ≥1 aeroallergen**	12 (63.2)	16 (41)	28 (48.3)	0.19

* *p*-value < 0.05 = statistically significant. † Blood eosinophil count (131 tests [69.3%] in total). ‡ Aeroallergen sensitization (58 tests [30.9%] in total).

## Data Availability

Not applicable.

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
