# Peer review of "Persistent Asthma at School Age: Associated Factors in Preschool Children with Asthma"

_children, 2022, doi:10.3390/children10010033_

Round 1

Reviewer 1 Report

Hello,

Thank you for submitting the manuscript titled “Persistent asthma at school age: Associated factors in preschool children with asthma”. The study suggested two modifiable factors that can be considered to lower the asthma severity in pre school to school age child transition phase.

The study conducted with various factors analysis vs characteristics, where breast feeding, housed in industrial area and co morbidities are stand out trait seen in children for persistent asthma.

There are further suggestions that authors should consider for the revision.

Minor:

1.       The literature review is not comprehensive and should include more relevant information when more studies done in children and asthma.

2.       Certainly, there are immunology aspects that children lack or develops as they grow, can you discuss the Th1 and Th2 cell role in this.

3.       The study would have stronger impact if there were some innate immunity and cell mediated immunity data with blood/plasma analysis in these children. This can be discussed in next plan and potential outcome.

Author Response

Response to Reviewer 1 Comments

Point 1: The literature review is not comprehensive and should include more relevant information when more studies done in children and asthma.

Response 1: Thank you for your suggestion. We have further review updated relevant information into the introduction part. [paragraph 2, line 48-66] Accordingly, persistence asthma in childhood is driven by gene-environmental interaction (e.g., genetic, epigenetic, environment). Multifaceted modifiable environmental risk factors play a crucial role in the preventive strategies, but the essential components have not clearly determined. A focusing on allergen sensitization, pollutants, and nutrition may be influenced in the persistent asthma symptoms in later childhood. The current strategies were decreasing risk of recurrent wheezing in early life among breastfeeding and maintain sufficient level of vitamin D during pregnancy through delivery. These protective factors may reduce wheezing episodes in early life, but the evidence of a decrease in the risk of persistent asthma at the age 6 years remains unclear. Sensitization to common aeroallergens (e.g., house dust mite, indoor pets, mold in home environment) appear to be more complex relation between indoor exposure and asthma outcome. A meta-analysis found intervention designed to prevent exposure to a single aeroallergen did not significantly affect asthma development, but multifaceted interventions of aeroallergen avoidance demonstrated an importance protective effect both before and after age of 6 years. Additionally, maternal smoking during pregnancy and after birth had its strongest effect on young children, whereas environmental tobacco smoke exposure seemed relevant to asthma development in older children. Furthermore, exposure to outdoor pollutants such as traffic-related pollution, living in an industrial area may be attributable to increased risk of asthma at any age.

Point 2: Certainly, there are immunology aspects that children lack or develop as they grow, can you discuss the Th1 and Th2 cell role in this

Response 2: Thank you. This suggestion is immensely helpful. We have emphasized this point in the discussion section. [paragraph 2, line 206-217]   Two major phenotypes of childhood asthma classified into: non allergic asthma (Th1-predominant) and allergic asthma (Th2-predominant). Allergic asthma is manifested by allergen sensitization to specific allergen, high immunoglobulin (Ig) E level, eosinophilia, and more prevalent during childhood. In early life the immune system is mainly response by a component of innate immunity. Although cellular innate immunity is present, its function of antigen presentation and activate specific T cell and B cell response, are not yet fully development. Regarding weak function of adaptive immune system, the young children generally expressed Th2 response. The immune regulation includes balanced innate immunity (e.g., via innate lymphoid cells) and equilibrium of T cell subpopulation (e.g., via regulatory T cell) to counter-regulate potential pro-inflammatory cytokines. Regulatory T cell (Treg) play a crucial role in a proper balance between Th1 and Th2, there function is different depend on maturity stages of the immune system. The immune dysregulation between Th1 and Th2 after birth leading to a part of pathogenesis of childhood asthma. Early-onset childhood asthma (allergic asthma) has been characterized by a Th2 shifted endotype and decrease innate immunity gene expression. External prenatal influences by maternal allergen exposure during pregnancy result in stronger Th-2 biased response. This response reinforced by continuous exposure to same antigens in the postnatal period leading to persistent IgE response. A prolonged shift toward Th2 immunity is associated with an increased risk for sensitization. This can result in airway inflammation at a sensible stage during lung development. Recognized modification caused during pre- and post-natal development possible preventive strategies specifically for children at risk.

Point 3: The study would have stronger impact if there were some innate immunity and cell mediated immunity data with blood/plasma analysis in these children. This can be discussed in next plan and potential outcome.

Response 3: Thank you for pointing this out. We agree with your comment. We have stated this as our limitation and mentioned to the potential variables for the next plan in the discussion section. [mentioned in paragraph 4, line 271-273]

Reviewer 2 Report

Asthma is a disease with an unpredictable history, so it is not possible to know in every child what its future evolution will be. However, there is evidence that if it is not treated appropriately and if the diagnosis and treatment plan is delayed for a long time, the symptoms tend to become more frequent. Therefore, this article is relevant and current when, it identifies in preschool age, factors that may be responsible for the worsening of asthma in school age and proposes strategies to minimize them.

The article under review is clearly written and well structured. Its experimental design was appropriate keeping in mind the objective of the study.

The results are clearly presented in appropriate tables and figures that clearly show the data collected, which facilitates their interpretation and discussion.

The conclusions presented are consistent with the discussion of the results and are very relevant because they add to the existing knowledge on the subject.

The references used in the introduction and discussion chapters, although few in number, are recent, relevant and sufficient publications.

It is suggested that future studies should include spirometry as a study variable.

For all of the above, the authors of the article are to be congratulated!

Author Response

Response of reviewer 2's comment

Point: It is suggested that future studies should include spirometry as a study variable.

Response: we appreciate the time and effort you have dedicated to providing valuable feedback on our manuscript. We have incorporated the suggestion through our manuscript. We have stated this as our limitation and mentioned to the potential variables for the next plan in the discussion section. [mentioned in paragraph 4, line 271-273]